# STL-Drive: Formal Verification Guided End-to-end Automated Driving

## ABSTRACT

End-to-end automated driving behavior models require extensive training data from machine or human driver experts or interacting with the environment to learn a driving policy. Not all human driver expert data represent safe driving that the end-to-end model is learning to imitate, and similarly, neither are some of the behaviors learned during exploration while learning by trial and error. However, the models should learn from such data without being negatively affected during the learning process. We aim to provide a learning framework to incorporate formal verification methods to improve the robustness and safety of the learned models in the presence of training data that contain unsafe behaviors, dubbed as STL-Drive. We are particularly interested in utilizing this framework to enhance the safety of end-to-end automated driving models. In this work, we incorporate Signal Temporal Logic (STL) as the formal method to impose safety constraints. In addition, we utilize the Responsibility-Sensitive Safety (RSS) framework to define the safety constraints. We designed a loss function that combines the task objectives and the STL robustness score to balance the learned policy's performance and safety. We demonstrate that encoding safety constraints using STL and utilizing the robustness score during training improves the performance and safety of the driving policy. We validate our framework using open-loop predictive simulator NAVSIM and real-world data from OpenScene. The results of this study suggest a promising research direction where formal methods can enhance the safety and resilience of deep learning models. Formal verification of safety constraints for automated driving will further increase the public's trust in automated vehicles.

## 1 INTRODUCTION

In recent years, automated driving systems have seen remarkable advancements, transforming how we envision the future of transportation. These systems, whether semi-automated or fully automated, are designed to enhance human mobility by making travel safer, more efficient, and more convenient. However, their value lies in meeting these mobility needs and ensuring the highest safety standards for all road users, including human drivers and pedestrians. Automated vehicles must prioritize accident prevention in critical scenarios, even compensating for other drivers' mistakes, to create a safer, more reliable transportation ecosystem. For example, in Figure- 1, we observe different agents reacting to the same scenario. The green dots represent the human driver's trajectory, while the red dots represent the learned model's trajectory. The automated policy on the left mimics the expert trajectory as closely as possible. At the same time, the robustness-aware RSS-Trained model on the right adopts a more conservative approach by slowing down since there is a vehicle traveling in the same lane that the ego-vehicle is attempting to merge with. In this case, the Transfuser model's prediction of future waypoints closely matches the expert trajectory. However, it is not necessarily safe since the human driver can drive unsafely.

The modern approach to automated driving is formulating the task as a learning problem where the agent has to learn a driving policy from data. Imitation Learning and Reinforcement Learning are the two most popular learning methods in automated driving literature. Imitation learning (IL) (Chen et al., 2020a; Chen & Krähenbühl, 2022; Prakash et al., 2021) is a form of supervised learning where the agent learns a policy to mimic the behavior of an expert. Expert demonstrations from Human drivers or an automated vehicle with access to the states of other agents in a simulator are collected.

IL methods suffer from distribution shift, i.e., if the agent encounters an unseen situation not present in the training data, the agent will fail to take appropriate action and continue to do so without recovering.

On the other hand, Deep Reinforcement Learning (DRL) (Toromanoff et al., 2020; Chen et al., 2020a; Chekroun et al., 2021) methods are more robust to distribution shift than IL by allowing the agent to interact with the environment to learn a policy by trial-and-error. At every step, the agent takes an action and receives a reward from the environment, which signifies how good the action is. The agent's goal is to maximize the sum of the accumulated rewards, thereby learning a sequence of actions that achieve the highest reward. Since the agent learns independently by exploring the environment using only the reward function, DRL methods require more data than IL methods to converge to an optimal policy. When collecting data through trial and error is expensive, researchers use offline RL methods such as (Kumar et al., 2020) to learn a policy from collected data. These methods are akin to supervised learning, where the agent learns to map input states to action pairs.


<div>Transfuser</div>
<div>RSS-Trained Transfuser</div>
</div>

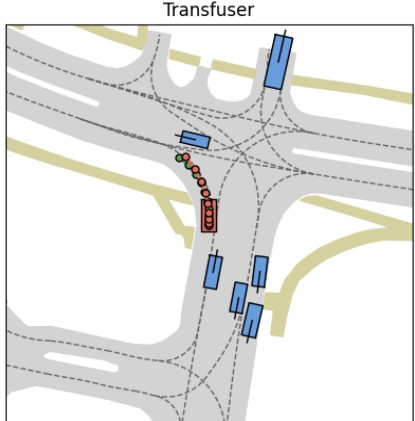 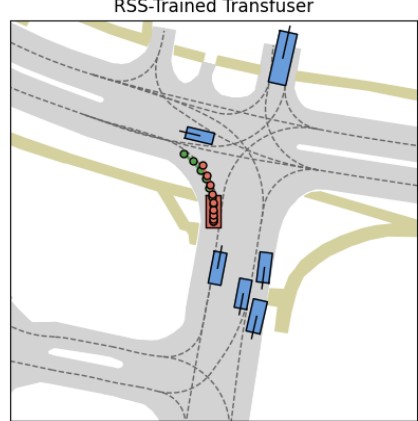

Figure 1: In this scenario, two different agents demonstrate trajectory prediction. The green dots represent the human driver's trajectory, while the red dots represent the learned model's trajectory. The training of the Transfuser model occurs without robustness loss while training the RSS-Trained Transfuser agent involves both robustness loss and task-related loss.

To validate the safety of vehicle interactions on the road, researchers have proposed driving safety assessment metrics (Wishart et al., 2020), which provide formal definitions to measure the safety of driving behavior. The Responsibility-Sensitive Safety (RSS) framework (Shalev-Shwartz et al., 2017) also calculates the minimum safe distance a pair of vehicles should maintain for a safe interaction in both lateral and longitudinal cases. When a violation of this minimum safe distance occurs, the RSS framework provides a proper response action for the ego-vehicle to restore the minimum distance to a safe distance again. RSS is quickly becoming a safety standard and has been used to improve adaptive cruise control (ACC) (Brosowsky et al., 2021), lane change behaviors (Naumann et al., 2019) for automated driving tasks. RSS has also been used for safety testing and validation of Automated Vehicle (AV) behaviors using falsification techniques in (Hekmatnejad et al., 2020) to identify the scenarios where an AV might become unsafe. Despite these efforts, research has been lacking in utilizing RSS to learn a driving policy for end-to-end automated driving. Our work STL-Drive fills this critical gap.

Our contributions to this work are as follows:

- Using RSS framework to train a robust and safe policy rather than using it as a real-time (reactive) safety monitor;
- Using Signal Temporal Logic Robustness score as a loss term in imitation learning;
- Finally, we formulate STL-Drive, which adopts Imitation learning to combine the robustness loss to improve the safety and robustness of a learned automated driving policy.

## 2 METHODOLOGY

In this work, we formulate AV decision-making as a waypoint prediction problem using Imitation learning. Instead of predicting low-level control signals, we predict the waypoints the AV will take in the future. To improve the robustness and safety of the automated driving policy, we use the RSS model as safety constraints to enforce the minimum safe distance measure. Using STL, we encode the RSS minimum safe distance rules and compute the robustness score to verify whether the vehicle maintains the minimum safe distance. We use the robustness score as an additional loss term with IL task loss to improve the performance and safety of the automated driving policy. We test and validate our hypothesis on the NAVSIM benchmark.

### 2.1 FORMULATION

**Imitation Learning:** Imitation Learning (IL) is a form of supervised learning where the task is to learn a policy $\pi$ that imitates an expert policy $\pi^*$. The policy $\pi$ maps the sensory inputs to waypoints in the Bird's Eye View (BEV) coordinates centered on the ego-vehicle. Given a list of expert waypoints, $\mathcal{W}$, where $\mathcal{W} = \{\mathbf{w}_t = (x_t, y_t)\}_{t=1}^{T}$. The goal is to learn a policy $\pi$ that minimizes the loss function $\mathcal{L}_{tpp}$;

$$\arg \min_{\pi} \mathbb{E}_{(\mathcal{X}, \mathcal{W}) \sim D} \left[ \mathcal{L}_{tpp}(\mathcal{W}, \pi(\mathcal{X})) \right], \tag{1}$$

where $D = \left\{ \left( \mathcal{X}^i, \mathcal{W}^i \right) \right\}_{i=1}^{N}$ of size $N$ is the dataset of all expert logs.

**Signal Temporal Logic:** (STL) (Maler & Ničković, 2004) is a formalism used for specifying and reasoning about temporal properties of signals in cyber-physical systems. STL extends classical temporal logic by introducing quantitative predicates over real-valued signals, allowing for the expression of timing constraints and conditions. STL is particularly useful for specifying behaviors such as safety, liveness, and performance requirements in systems where time plays a critical role. The robustness score evaluates how well a system adheres to the specified temporal properties. Unlike traditional Boolean evaluations that provide binary outcomes (true/false), robustness scores offer a scalar value indicating the degree of satisfaction or violation of the STL specifications. This not only helps with the testing of the system capabilities using falsifying techniques but also for the real-time monitoring of systems, which is valuable in applications requiring high precision and reliability, such as automated control systems and safety-critical monitoring.

**Responsibility-Sensitive Safety:** The RSS (Shalev-Shwartz et al., 2017) model is a formal framework designed to ensure the safety of automated driving systems. It provides a set of guidelines and rules that define what constitutes safe driving behavior under various traffic scenarios. RSS model ensures that vehicles maintain a safe distance by considering speed and reaction time, dictates appropriate responses to other vehicles' actions like braking or changing lanes, and outlines appropriate actions for handling dangerous situations to avoid collisions. It uses mathematical and logical formulations to provide formal safety guarantees, ensuring predictable system behavior. Safety distance formulation by the framework considers worst-case scenarios to establish safety boundaries that account for human error and unpredictable events. In this work, we encode the Minimum Safe Distance requirements in STL as follows:

$$D^{lat} \equiv (d_{lat} \geq d_{min,lat})$$
$$D^{lon} \equiv (d_{lon} \geq d_{min,lon})$$
$$D^{lat,lon} \equiv D^{lat} \wedge D^{lon}$$
$$S^{lat} \equiv \Box(D^{lat})$$
$$S^{lon} \equiv \Box(D^{lon})$$
$$S^{lat,lon} \equiv \Box(D^{lat,lon})$$

For RSS model parameters, we consider the RSS CITS parameter set from (Wishart et al., 2020) as shown in Table-1 but with a slower response time ($\tilde{\rho}$) of 0.5 seconds to compute the Minimum Safety Envelope for longitudinal and lateral scenarios.

| Parameter | Value | Parameter | Value |
|---|---|---|---|
| $a_{maxAcc}^{lon}$ | $1.8\ m/s^2$ | $a_{maxBr}^{lon}$ | $6.1\ m/s^2$ |
| $a_{minBr}^{lon}$ | $3.6\ m/s^2$ | $\tilde{\rho}$ | $0.5$ s |
| $a_{minAcc}^{lat}$ | $5.88\ m/s^2$ | $a_{maxAcc}^{lat}$ | $8.83\ m/s^2$ |

Table 1: Longitudinal and Lateral Acceleration and Braking Parameters

where, $d_{lat}$ is the lateral distance between two vehicles and $d_{min,lat}$ is the minimum safety envelope in the lateral direction computed using RSS formulation. $d_{lon}$ is the longitudinal bumper-to-bumper distance between two vehicles and $d_{min,lon}$ is the minimum safety envelope in the longitudinal direction. The above rules, specifically $D^{lat}, D^{lon}$ and $D^{lat,lon}$, are boolean predicates that check if the minimum safety envelope is maintained or not. The rules $S^{lat}, S^{lon}$ and $S^{lat,lon}$ have an always ($\square$) operator which checks for satisfaction for the entire interaction the vehicles are involved in. Combining these approaches, the above-specified requirements are encoded using RTAMT (Ničković & Yamaguchi, 2020) monitoring tool to compute the robustness score.

## 2.2 ROBUSTNESS SCORE

In the context of RSS, safety envelope distances are calculated for pairs of vehicles. So, at every time step, robustness is calculated for only one other vehicle and the ego-vehicle. So, we consider two variants of robustness scoring: (1) where the minimum robustness score is chosen at each time step (Type-0) from all the pairs of vehicles that are less than $50.0\ meters$ apart from the ego-vehicle and (2) where the robustness score of only the closest vehicle is considered (Type-1). However, in reality, the ego-vehicle interacts with multiple vehicles simultaneously. Therefore, the robustness score at each time step should also consider the influence of other vehicles or pedestrians. To consider this effect, we use the inverse weighted distance average to combine the robustness scores to account for the proximity of other agents to the ego-vehicle. This is the third variant (Type-2) we consider for calculating robustness. The combined robustness score at time step $t$ for each type is defined as follows:

**Type-0:**

$$\rho(t) = \min_{i \in \mathcal{N}(t)} \rho(i), \tag{2}$$

where $\mathcal{N}(t)$ represents the set of all vehicles that are less than $50.0\ meters$ from the ego-vehicle at time step $t$, and $\rho(i)$ is the robustness score of the vehicle $i$.

**Type-1:**

$$\rho(t) = \rho(i^*), \tag{3}$$

where $i^*$ is the closest vehicle to the ego-vehicle at time step $t$, and $\rho(i^*)$ is the robustness score of this closest vehicle.

**Type-2:**

$$\rho(t) = \frac{\sum_i w_i \cdot \rho(i)}{\sum_i w_i}, \tag{4}$$

where $w_i$ is equal to $1/d_i$ and $d_i$ is the distance of the vehicle $i$ from the ego-vehicle and $\rho(i)$ is the robustness score of the vehicle $i$. After obtaining the combined robustness score for each time step, we compute the robustness score for the entire trace ($\tau$), $\rho_\tau$, equal to the minimum robustness score across the entire time duration.

$$\rho_\tau = \min_{t \in [0,\tau]} \rho(t), \tag{5}$$

For training the Imitation Learning agent, which is robustly guided by safety constraints that are encoded in Signal Temporal Logic, we modify the original loss function of the Transfuser (Chitta et al., 2023) model by adding an additional loss term that controls the robustness weight denoted by $\alpha$ ($\alpha \in [0, 1]$). Let $\mathcal{L}_{tpp}$ be the loss function of the Transfuser model, which is a combination of trajectory loss, semantic segmentation, bounding box, and additional semantic loss in the BEV

frame. We combine this task loss function with robustness loss in a weighted fashion as follows:

$$\mathcal{L}_\alpha = (1 - \alpha) \cdot \mathcal{L}_{tpp} + \alpha \cdot \frac{1}{\|D\|} \sum_{i=1}^{\|D\|} \rho_\tau^{(i)}. \tag{6}$$

This final loss function $\mathcal{L}_\alpha$ trains the agents to mimic the expert driving behavior while being robust and safe.

## 3  EXPERIMENTAL RESULTS

### 3.1  DATASET & TRAINING

To train and evaluate our models on real-world data, we use the OpenScene (Contributors, 2023) dataset, an extension of the nuPlan (Caesar et al., 2021) dataset developed for autonomous vehicle research. OpenScene focuses on understanding 3D dynamic scenes for autonomous driving and perception tasks like object detection, 3D scene understanding, and semantic segmentation. The nuPlan dataset contains approximately 1200h of driving data from 4 cities: Boston, Pittsburgh, Las Vegas, and Singapore. About 838h are from Las Vegas, with the remaining data split equally across the other cities. To train, validate, and test our learned models, we use the NAVSIM benchmark (Dauner et al., 2024). NAVSIM is a simulation-based benchmark developed for large-scale, data-driven evaluation of autonomous driving systems. It enables testing autonomous vehicles in realistic environments using a non-reactive open-loop simulation approach. In diverse driving scenarios, NAVSIM evaluates driving policies on critical aspects like safety, comfort, and navigation progress as reported in Table-2. It also gives the flexibility to use other real-world datasets.

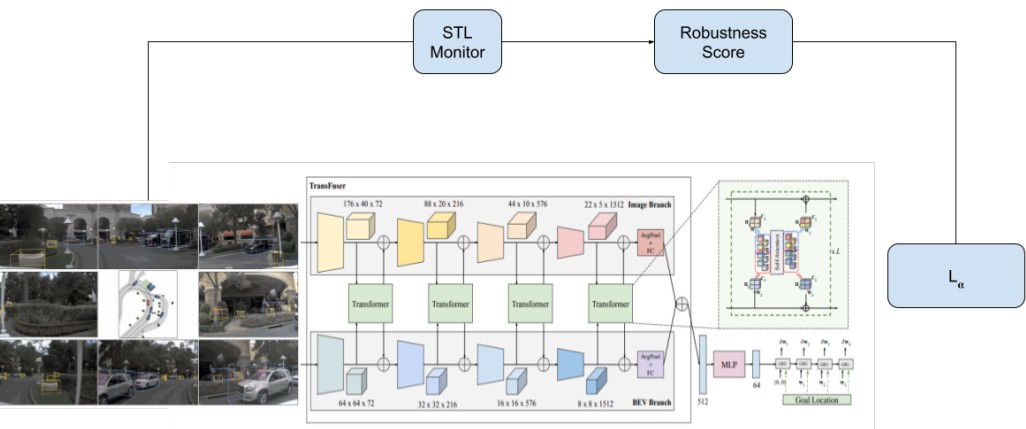

Figure 2: Overview of the transfuser architecture and STL-Drive

**Training Details:** For the baseline, we chose Transfuser (Chitta et al., 2023), one of the top-performing Imitation Learning algorithms on CARLA Leaderboard-v1.0. Transfuser combines contextual and spatial information from sensors like camera and LIDAR using a Transformer and learns an automated driving policy that handles the driving task and safety well. We use an implementation of the Transfuser provided by the NAVSIM benchmark. To test our hypothesis, we trained each model on the train split (*train_test_split=navtrain*), which contains 103288 scenarios interpolated at 10Hz with a duration of 4.0 seconds each. Similarly, we evaluate all the trained models using the test split (*train_test_split=navtest*).

To understand the significance of $\alpha$, we trained the models with the following values: $[0.2, 0.5, 0.8, 1.0]$. To test how the robustness type formulation affects the trained models, for Type-1 and Type-2 robustness types, we train the models with $\alpha = 0.5$. We also demonstrate the significance of using the RSS framework's minimum safety spatial envelope as a safety constraint. We use a static safety lateral and longitudinal distance envelope of $0.5\ meters$ for the minimum safety spatial envelope comparison.

## 3.2 RESULTS ANALYSIS

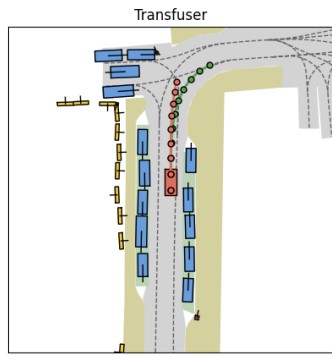 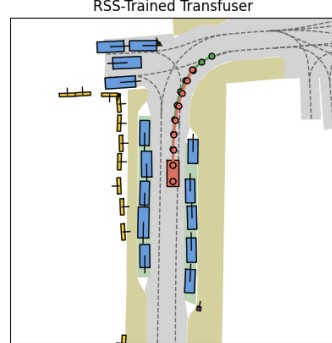

(a) Scenario 1: STL-Drive model following the expert trajectory.

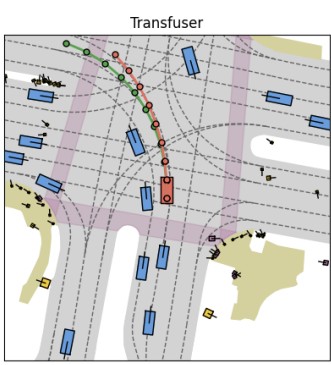 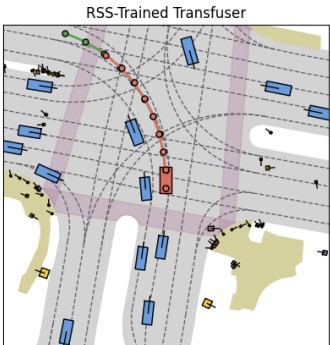

(b) Scenario 2: STL-Drive model following the expert trajectory and traffic rules inherently.

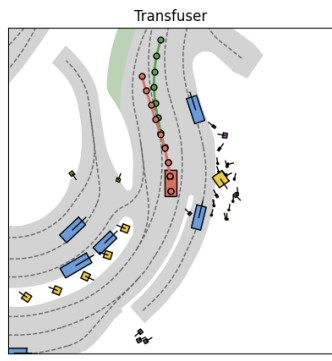 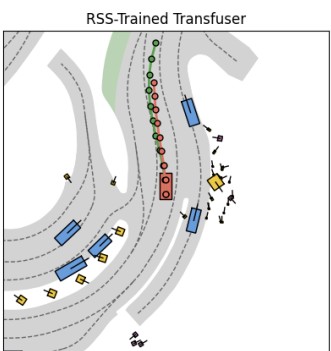

(c) Scenario 3: STL-Drive model following the expert trajectory on a curved road.

Figure 3: Various scenarios showcasing how the Robustness Aware learner exhibits safe behaviors. The green dots represent the human driver's trajectory, while the red dots represent the learned model's trajectory.

From Table-2, we observe the evaluation results of different policies. The model with $\alpha = 0$ is the baseline Transfuser model with the lowest performing score of 0.7409. All STL monitoring-based agents have higher scores than the baseline agent suggesting that encoding safety constraints

in Signal Temporal Logic and using the robustness score can make the learned policies more robust and safe. For the variation of robustness type, we observe that Type-1 has the best performance amongst other formulations of Type-0 and Type-2 suggesting that the closest vehicle to the ego vehicle will influence the vehicle's safety more than the other vehicles in the scenario.

| Monitor Type | Robustness Type | $\alpha$ | No At-Fault Collisions ↑ | Drivable Area Compliance ↑ | Driving Direction Compliance ↑ | Ego Progress ↑ | Time to Collision Within Bound ↑ | Comfort ↑ | Score ↑ |
|---|---|---|---|---|---|---|---|---|---|
| - | - | 0 | 0.9584 | 0.8389 | 0.6921 | **0.9027** | **0.9999** | 0.9608 | 0.7409 |
| RSS Monitor | 0 | 0.2 | **0.9646** | 0.8618 | **0.9687** | 0.7166 | **0.9060** | **0.9999** | 0.7648 |
|  |  | 0.5 | 0.9609 | **0.8671** | 0.9606 | **0.7255** | 0.8997 | **0.9999** | **0.7674** |
|  |  | 0.8 | 0.9632 | 0.8547 | 0.9600 | 0.7173 | 0.9008 | **0.9999** | 0.7575 |
|  |  | 1.0 | 0.7388 | 0.5696 | 0.8886 | 0.1178 | 0.7061 | 0.3560 | 0.2685 |
|  | 1 | 0.5 | 0.9702 | 0.8785 | 0.7372 | 0.9132 | **0.9999** | 0.9640 | 0.7837 |
|  | 2 | 0.5 | 0.9620 | 0.8545 | 0.7194 | 0.8886 | **0.9999** | 0.9576 | 0.7547 |
| Constant Distance Envelope | 0 | 0.2 | 0.9609 | 0.8587 | 0.9626 | 0.7174 | 0.8930 | **0.9999** | 0.7575 |
|  |  | 0.5 | 0.9552 | 0.8380 | 0.9567 | 0.6942 | 0.8899 | **0.9999** | 0.7356 |

Table 2: Table of Performance Metrics for Different Monitor Types and $\alpha$ Values

Table 2 shows the importance of RSS minimum safety spatial envelope compared to a constant distance minimum safety spatial envelope. As the value of $\alpha$ increases, the performance drops significantly for the constant distance envelope monitor. When comparing the results for the same value of $\alpha$ across both envelopes, the RSS minimum safety spatial envelope performs better than the constant distance safety spatial envelope, suggesting that the RSS framework is the better choice for minimum safety spatial envelope.

Now, we look at some scenarios presented in Figure-3 to see how robustness can improve the safety of the automated policy. Each row corresponds to the same scenario and the left image shows the future trajectory predictions from the Transfuser agent, while the right image corresponds to the agent trained with robustness formulation. In Scenario-1, the baseline ($\alpha = 0$) model does not mimic the expert trajectory, whereas the STL-Drive model follows a similar trajectory to the expert log. When multiple vehicles are present, the task loss, which consists of bounding boxes and localization of other vehicles, can cause it to neglect the safety of the policy. In Scenario 2, even though both agents mimic the expected trajectory, the STL-Drive model follows traffic rules and takes the leftmost possible trajectory to merge into the lane. In Scenario 3, we again observe that the STL-Drive model outperforms the $\alpha = 0$ policy on roads with complex driving behaviors. These scenarios support our hypothesis that the robustness-aware automated driving policy is more robust and safe than the policy with $\alpha = 0$. The above results demonstrate that having a robustness framework that considers safety constraints while training will improve the robustness and safety of the learned policy.

## 4 RELATED WORK AND BACKGROUND

**End-to-End Automated Driving with Imitation Learning:** Since RL relies on trial and error, it is unsafe, costly, and data-inefficient to train automated driving vehicles in the physical world. Hence, most of the research related to automated driving is studied in simulation. For example, CARLA (Dosovitskiy et al., 2017) is a popular choice for simulating realistic urban traffic interactions that include intersections, multiple vehicles, pedestrians, traffic lights, and stop signs, along with weather conditions affecting visibility. In CARLA, the task is to learn a goal-based navigation policy given a list of waypoints and high-level commands. Since there is a gap between simulated and real-world data, some methods utilize recorded expert human driver logs to learn a policy using Imitation Learning. In (Chekroun et al., 2021), the authors propose combining Imitation Learning and Reinforcement Learning to utilize expert demonstrations and exploration to solve the distribution shift of general IL methods and sample inefficiency of RL methods. (Chitta et al., 2023) predict waypoints using Imitation Learning by using sensor fusion and a transformer to gather spatial and temporal context. (Chen & Krähenbühl, 2022) similarly predicts waypoints by learning to predict the trajectories of other agents in the scene.

**Safe Imitation Learning:** Constrained Imitation Learning, focuses on replicating expert driving behavior while adhering to safety constraints. In Constrained Behavioral Cloning, the vehicle learns to imitate expert trajectories but avoids unsafe actions like speeding or tailgating, ensuring a balance between imitation and safety (Bojarski et al., 2016). Enhancements like Safe DAgger improve upon this by incorporating safety checks during the training process, preventing the model from learning unsafe behaviors even as it adapts and refines its policy (Ross et al., 2011). Additionally, real-time mechanisms such as Model Predictive Control (MPC) Shields monitor the vehicle's predicted trajectory, overriding any potentially dangerous actions to prevent accidents or collisions (Chen et al., 2020b). In contrast, we use the safety constraints offline to learn a robust policy without the need to constrain the policy in real time.

Safe Inverse Reinforcement Learning (IRL) methods solve this issue by inferring the reward functions that drive human behavior, integrating safety into the learning process. Safe IRL helps vehicles infer typical driving objectives, such as efficiency and safety while penalizing unsafe actions like running red lights or following too closely (Ziebart et al., 2008). Risk-sensitive IRL introduces additional caution in more high-risk environments, allowing autonomous systems to make conservative decisions in complex scenarios like dense urban traffic or interactions with vulnerable road users, such as pedestrians (Levine et al., 2011). However, inferring the reward function preferred by a human driver is a complex task and would need a lot of feature engineering to search for the optimal reward function that can improve the policy. Our method does not require such a task of finding an appropriate reward function.

End-to-end learning with Safety Enhancements focuses on directly mapping sensor inputs, such as camera or LiDAR data, to driving control actions. These systems require careful regularization to avoid unsafe behavior. For example, barrier function-based safety regularization penalizes unsafe trajectories, keeping the vehicle within safe zones (Gurriet et al., 2018). Learning from safe demonstrations ensures that models only learn from optimal, expert-level behavior by filtering out unsafe or sub-optimal human driving data (Zhang & Cho, 2016). Additionally, interactive imitation learning allows real-time expert intervention to correct unsafe actions during training (Ross et al., 2011). Hybrid approaches like Safe RL, on top of Imitation, combine imitation learning with reinforcement learning, allowing the vehicle to learn from demonstrations and then refine its behavior through RL, ensuring both performance and safety in complex scenarios like merging onto highways or navigating intersections (Kahn et al., 2017). Our work follows a similar approach by using the RSS framework to guide and train the automated policy.

## 5 CONCLUSION

In this work, we propose a framework to combine task-based loss function with robustness loss function to learn a safe and robust automated driving policy. RSS rules are encoded using Signal Temporal Logic, and given a trace of the vehicle behavior, we compute the robustness score and utilize this robustness score in a loss function. We trained an imitation learning agent using the new loss function that was shown to outperform a similar driving policy without the robustness loss.

The rapid evolution of artificial intelligence, especially in the realm of generative models, mirrors the multifaceted nature of human intelligence. While verbal intelligence has driven early advances through large language models (LLMs) and text-prompted generative AI, the future of AI lies in developing spatial intelligence. This shift is critical in fields such as automated driving, where understanding and interacting with the physical world is paramount. In the context of automated driving, as discussed in this work, spatial intelligence manifests in the ability of AI models to reason about and respond to complex, real-world scenarios involving dynamic environments and safety constraints. The integration of Signal Temporal Logic (STL) and the Responsibility-Sensitive Safety (RSS) framework enhances the spatial reasoning capabilities of AI, enabling safer and more reliable decision-making. As we move forward, developing AI systems with deeper spatial understanding will be the key to unlocking the next phase of human-machine collaboration, allowing us to build smarter vehicles and better, safer environments.

## 6 LIMITATIONS

While our framework STL-Drive demonstrates advancements in integrating formal verification methods, such as STL and the RSS framework, to enhance the safety of end-to-end automated driving systems, it is not without limitations. The first limitation lies in the dependency on the quality and quantity of the training data. The scope of the training scenarios directly influences the performance of our model, and although we incorporated diverse datasets like OpenScene, there may still be unaccounted edge cases in real-world environments. Additionally, while incorporating STL-based safety constraints offers a rigorous approach to improving robustness, the reliance on predefined logical safety rules may limit flexibility in novel, unpredictable driving scenarios. Our model has been designed to handle various situations, yet unforeseen conditions not covered by the defined temporal logic could challenge the framework's adaptability. Another limitation is the computational cost associated with calculating robustness scores during training. While using the RSS model for enforcing safety has improved results, these computations can be resource-intensive, making it challenging to scale for real-time applications in highly dynamic environments. It is worth highlighting that this is a trade-off between the depth of safety verification and the speed of decision-making, which could impact the model's deployment in real-world systems.

Despite these constraints, we believe STL-Drive's contributions to the field are substantial. By incorporating formal verification into the training process, we have introduced a new pathway for enhancing the safety and reliability of automated driving policies (and other safety-critical real-world applications, such as manufacturing procedures and tactical logistics), laying the groundwork for further exploration in combining machine learning algorithms (system 1 thnking) with formal safety guarantees (system 2 thinking). STL-Drive represents a promising step toward safer, more robust automated driving systems that can inspire continued research and development in this domain.

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
