# OpenReview forum: "STL-Drive: Formal Verification Guided End-to-end Automated Driving"
_ICLR.cc/2025/Conference — ICLR 2025 Conference Withdrawn Submission_

### Official Review · Reviewer_9dh3 · 2024-10-20

**Soundness:** 2
**Presentation:** 1
**Contribution:** 1
**Rating:** 1
**Confidence:** 4

**Summary:**

The paper focuses on reinforcement learning with STL constraints. The Responsibility-Sensitive Safety model is described by STL specification and the robustness score is defined to provide a safety loss in RL.

**Strengths:**

This paper considers the safety of autonomous driving which is an important problem.

**Weaknesses:**

There are several major issues in writing and novelty.

1. Why do the authors encode the minimum safety distance requirement into STL specifications? First, there are no real-time constraints in the specifications given in this paper. Thus, they are actually LTL rather than STL. In fact, these can be simply modeled as propositional logic specifications, where the equations in Lines 152, 153, and 154 are already sufficient. Please clarify why they have to be described as STL.
2. Why do the authors define the new robustness score? The robustness of a trajectory in terms of an STL specification was proposed a decade ago [1] and has been extensively studied [2]. Please clarify the motivation of proposing a new definition of robustness here.
3. Encoding STL robustness into loss function for RL has also been widely discussed since 2020 [2,3,4]. This paper does not discuss these works either theoretically or empirically, which raises a major concern about the contribution of this work. Please clarify the main differences and contributions compared with these works.


[1] Donzé, Alexandre, Thomas Ferrere, and Oded Maler. "Efficient robust monitoring for STL." In Computer Aided Verification: 25th International Conference, CAV 2013, Saint Petersburg, Russia, July 13-19, 2013. Proceedings 25, pp. 264-279. Springer Berlin Heidelberg, 2013.

[2] K. Leung, N. Aréchiga, and M. Pavone, "Back-propagation through STL specifications: Infusing logical structure into gradient-based methods," in Workshop on Algorithmic Foundations of Robotics, Oulu, Finland, 2020.

[3] Kapoor, Parv, Anand Balakrishnan, and Jyotirmoy V. Deshmukh. "Model-based reinforcement learning from signal temporal logic specifications." arXiv preprint arXiv:2011.04950 (2020).

[4] Kalagarla, Krishna C., Rahul Jain, and Pierluigi Nuzzo. "Model-free reinforcement learning for optimal control of Markov decision processes under signal temporal logic specifications." In 2021 60th IEEE Conference on Decision and Control (CDC), pp. 2252-2257. IEEE, 2021.

**Questions:**

Please see the weaknesses.

---

### Official Review · Reviewer_omF3 · 2024-10-27

**Soundness:** 2
**Presentation:** 2
**Contribution:** 2
**Rating:** 3
**Confidence:** 4

**Summary:**

This paper studies how to train a safe and robust policy for autonomous driving in the end-to-end fashion. Specifically, the responsibility sensitive safety (RSS) requirements are considered and encoded into signal temporal logic (STL) formula. Then, the authors use STL robustness value as another training loss term to encourage the trained policy to be robust as well. The authors discuss three types of robustness scores depending on what other vehicles are considered. Finally, the simulated results in NAVSIM with data from OpenScene help demonstrate the proposed framework.

**Strengths:**

1. I find the paper is generally well-presented and easy to follow.
2. It is meaningful to observe the ablation study w.r.t. alpha and different types of robustness score.

**Weaknesses:**

I am mainly concerned about the novelty/significance of this work. This paper mainly studies how to incorporate formal specifications into policy learning such that the trained driving policy can be robust and safe. And the authors use STL to encode RSS requirements so the STL robustness can serve as part of the objective function.

However, formal methods (e.g., STL, MTL, LTL) have been widely considered in policy learning to improve the safety and robustness, in both the imitation learning (IL) setting and reinforcement learning (RL) setting, see [1-3]. Note that some other approaches to optimize STL robustness scores rather than IL/RL methods have also been proposed extensively [4-5]. Thus, adding STL robustness term into the training loss and then formulate an IL training problem to learn the policy is not novel.
In addition, the integration of STL and the RSS framework is also not new, which was first proposed in [6].

The experimental validation part is also not very convincing from my perspective. For example, the evaluated metrics (e.g., comfort, score, ego progress, compliance) are not well specified and I am not sure what do these metrics mean. This is critical as the readers cannot understand the experimental significance of the approach if the metrics are not well-defined or cited from the literature. Also, the experiment is only conducted on one benchmark (NAVSIM) and compared with one baseline (TransFuser). I am concerned about its performance w.r.t. other baselines (e.g., [7], [8]) and generalizability to other benchmarks (e.g., the Longest6 Benchmark).

Please see more concerns and questions from my side in the "Questions" part.

[1] A formal methods approach to interpretable reinforcement learning for robotic planning. Science Robotics.

[2] Temporal Logic Specification-Conditioned Decision Transformer for Offline Safe Reinforcement Learning. arXiv preprint arXiv:2402.17217.

[3] Learning from demonstrations using signal temporal logic in stochastic and continuous domains. IEEE Robotics and Automation Letters, 6(4), pp.6250-6257.

[4] Backpropagation through signal temporal logic specifications: Infusing logical structure into gradient-based methods. The International Journal of Robotics Research, 42(6), pp.356-370.

[5] Smooth operator: Control using the smooth robustness of temporal logic. In 2017 IEEE Conference on Control Technology and Applications (CCTA) (pp. 1235-1240). IEEE.

[6] Encoding and monitoring responsibility sensitive safety rules for automated vehicles in signal temporal logic. In Proceedings of the 17th ACM-IEEE International Conference on Formal Methods and Models for System Design.

[7] Think twice before driving: Towards scalable decoders for end-to-end autonomous driving. In Proceedings of the IEEE/CVF Conference on Computer Vision and Pattern Recognition.

[8] Trajectory-guided control prediction for end-to-end autonomous driving: A simple yet strong baseline. Advances in Neural Information Processing Systems.

**Questions:**

1. I am a bit confused by the loss term (6): isn't it be (1-alpha)*L - alpha*robustness instead of (1-alpha)*L + alpha*robustness? My understanding is that the authors would like to minimize the task imitation loss and maximize the STL robustness score such that STL satisfaction is more robust.

2. I recommend that the authors should explain the evaluation metrics more formally such as comfort, score, compliance, etc. In the current version, the readers are unable to find the definitions/intuitions of these metrics, which prevent them from understanding the simulation results.

3. I would like to bring a relevant paper to the authors' attention, please see [6]. To my knowledge, [6] is the first paper to encode RSS specifications into STL formula. I think the authors did the similar thing in this paper, so I suggest that the authors can compare with [6] on the RSS-STL encoding techniques and have some discussions.

4. The Related Work section is too broad and the authors should dive deeper into the review of those papers on how to use formal methods to enable learning-based autonomous driving (or generally robotic systems) trustworthy, safe, and robust.

5. I would suggest the authors to formally introduce STL syntax and (quantitative) semantics before encoding RSS into STL. Currently formal definitions of syntax and semantic are missing, and those readers unfamiliar with STL may get confused when reading RSS-STL part.

6. The current experiment is only conducted on one benchmark and compared with one baseline. I suggest the authors to compare with more methods (like [7, 8]) and test on more benchmarks (e.g., Longest6 or Town05 Long).

7. The authors can use figures with higher resolutions (especially Figure 2 as it is the overall framework figure but not very clear visually).

---

### Official Review · Reviewer_Qfvu · 2024-10-31

**Soundness:** 2
**Presentation:** 2
**Contribution:** 1
**Rating:** 3
**Confidence:** 4

**Summary:**

This paper proposes a framework to improve the safety of end-to-end autonomous driving algorithms, in the presence of training data that may contain unsafe behavior. They use Responsibility-Sensitive Safety (RSS) model to encode a set of guidelines and rules for ensuring safety of the performance, specifically for maintaining a minimum safety distance between vehicles in longitudinal and lateral directions. Given the RSS safety guidelines, they use Signal Temporal Logic (STL) to formulate the rules and measure their robustness score to evaluate safety. The robustness values are included as an additional loss term into the loss function of TransFuser method, which is an imitation learning approach from literature, to learn safe behavior from expert trajectories. Their framework, called STL-Drive, is trained on real-world data from OpenScene and is evaluated on NAVSIM simulator. They showcase the effectiveness of their framework through multiple scenarios and compare the performance of the method by considering different weighted combination of the loss terms. The results confirm the advantage of including the robustness loss term in improving the performance of the car.

**Strengths:**

This paper raises up an interesting problem about ensuring safe learning from expert demonstrations, in an imitation learning framework. This is a significant concern as the data collection for specific tasks and scenarios can be expensive and all of the recorded traces in the existing datasets might not necessarily reflect safe behaviors, which makes the performance of agents trained by imitation learning prone to error. Using RSS safety metrics, formulated as STL formulas and imposed as a robustness loss term into the imitation learning's objective seems a valid idea, specifically for end-to-end autonomous driving. The method has been evaluated on real-world data from OpenScene dataset that is based on nuScene, which ensures the applicability of the suggested framework to real-world scenarios. In addition, the performance of the framework has been examined under various robustness weight $\alpha$, which clearly represents its importance in improving the performance of the learned trajectories. Overall, the paper is written well-organized, and the limitations of the framework have been discussed comprehensively.

**Weaknesses:**

Although this paper has raised a significant problem about safe imitation learning in autonomous driving, there are some drawbacks and missing points that need to be addressed:
1) In introduction, there are some arguments about imitation learning and RSS applications in automated driving; however, they don't explain what's the motivation for using STL formulas and how that is going to improve the performance of the framework, specifically in an imitation learning manner.
2) The relevant references included in the introduction and related works are mostly from 2020's, while there have been many works in the area of end-to-end autonomous driving over the last two-three years. In addition to that, the results have been compared only with respect to TransFuser baseline and there isn't any comparison with the latest works in this area. Although the idea of employing RSS in an end-to-end manner sounds interesting, the technical contributions of the proposed method are not well established.

Also, there are some minor issues about the structure of the paper:
1) In section 2.1 "Imitation Learning", the loss term $\mathcal{L}_{tpp}$ and set $\mathcal{X}$ are not defined.
2) In section 2.1 "Signal Temporal Logic", it could be better to include a very brief description of STL operators and its quantitative semantics as they are used in the next subsections. Also, a reference for the robustness degree of STL formulas is missing, for example:
Donzé, Alexandre, and Oded Maler. "Robust satisfaction of temporal logic over real-valued signals." International Conference on Formal Modeling and Analysis of Timed Systems. Berlin, Heidelberg: Springer Berlin Heidelberg, 2010.
3) In section 2.1 "Responsibility-Sensitive Safety", in the mathematical formulas representing the minimum safe distance requirements, the time bounds of the always operator $\square$ are missing.
4) In section 2.1 "Responsibility-Sensitive Safety", It's not specified where the parameters of Table 1 are used. In addition, the values of $d_{min, lat}$ and $d_{min,lon}$ are not reported. Also, I believe the paragraph starting with "where, $d_{lat}$ is the lateral distance ... " is misplaced and it should be right after the equations in this subsection.
5) In section 3.1, in Figure 2, the TransFuser architecture has very poor resolution and it's not readable.
6) In section 4, it could be better to include a subsection about application of formal methods, specifically STL formulas, in autonomous driving.

**Questions:**

1) Why STL-Drive is a promising framework for autonomous driving, compared to the state-of-the-art algorithms for end-to-end autonomous driving?

2) Besides the idea of employing RSS in an end-to-end driving framework, what are the major technical contributions of this paper?

---

### Official Review · Reviewer_uaU4 · 2024-11-02

**Soundness:** 3
**Presentation:** 1
**Contribution:** 1
**Rating:** 3
**Confidence:** 5

**Summary:**

This paper proposes to add a safety loss module for safer imitation learning for automated driving. The safety loss module is based on the robustness score from STL expression with RSS requirements.

**Strengths:**

The approach seems reasonable to me, as various existing works introduce STL robustness loss to enforce formal properties in training modules.

**Weaknesses:**

The approach is not novel at all. As stated above, it now becomes a very common way to introduce STL robustness score into some training modules, to enforce/optimize the corresponding formal properties.

The presentation is bad.
1. The paper is titled with formal verification guided end-to-end automated driving. This work only focuses on the trajectory planning/prediction side and does not cover perception, localization, low-level tracking control, etc. Therefore, the term 'end to end automated driving' is too big for this paper.
2. The writing and visualization could be improved a lot, for instance, I can barely see the difference in figure 1.

**Questions:**

Have you ever tried this as a baseline?
Filter out those "unsafe" data from the dataset by using the RSS rules and train the planning/prediction policy with the filtered data. How does it compare to your approach in the paper?

---

### Note · Authors · 2024-11-12

**Comment:**

I am withdrawing this submission. We thank the reviewers and the conference for the review and informative feedback.

**Withdrawal Confirmation:**

I have read and agree with the venue's withdrawal policy on behalf of myself and my co-authors.